# Exploring an Interdisciplinary Curriculum in Product and Media Design Education: Knowledge Innovation and Competency Development

Yi-Fang Kao [1], Hung-Cheng Chen [2],* and Jung-Hua Lo [3],*

1   Department of Product and Media Design, Fo Guang University, Yilan 26247, Taiwan; yfkao@mail.fgu.edu.tw
2   School of Education, Huanggang Normal University, Huanggang 438000, China
3   Department of Applied Informatics, Fo Guang University, Yilan 26247, Taiwan
*   Correspondence: hungcheng@hgnu.edu.cn (H.-C.C.); jhlo@mail.fgu.edu.tw (J.-H.L.)

**Abstract:** This study explores the implementation of an interdisciplinary curriculum in product and media design education and its impact on knowledge innovation and competency development. The curriculum is based on scaffolding theory, incorporating design proposals, workshops, digital design, marketing tests, and marketing activities guided by teachers and mentors from a branded ceramic workshop. The research findings demonstrate that the interdisciplinary curriculum facilitates knowledge innovation and fosters the development of students' professional skills, creativity, and problem-solving skills. The study also highlights the importance of stable scaffolding, including artefact-, peer-, and teacher support, which significantly contributes to cultivating transformational competencies, as outlined in the 2030 OECD Learning Compass. Additionally, the collaboration with Shanshing Four Seasons Celadon Studio on a patented ceramic product, "Funny Monkey", is a tangible example of the journey towards a triple-helix knowledge economy. This research underscores the importance of interdisciplinary curricula in promoting knowledge innovation and integrating transformational competencies in education. Scaffolding theory provides practical guidance for student learning and teaching strategies, presenting a sustainable roadmap for developing interdisciplinary curricula and offering a concrete and transferable pedagogical prototype for educational innovation.

**Keywords:** interdisciplinary curriculum; scaffolding theory; knowledge innovation; transformational competencies; triple-helix knowledge economy





## 1. Introduction

### 1.1. Research Background and Motivation

In the OECD Learning Compass 2030 report [1], the Organization for Economic Cooperation and Development (OECD) indicates a significant change in the global socioeconomic context. These changes include the growing need for schools to help students cope with rapidly changing environments and social changes, with jobs that have not yet been created, technologies that have not yet been invented, and with social problems that have not yet been anticipated. As a vital component of the OECD Learning Compass 2030 report, the concept of Knowledge for 2030 highlights four types of knowledge required to meet these challenges: disciplinary knowledge, interdisciplinary knowledge, epistemic knowledge, and procedural knowledge. Learning and constructing these four types of knowledge will meet the educational outlook and skill development of 2030.

The challenges presented in the OECD Learning Compass 2030 report also significantly impact many aspects of design education, such as exploring sustainable development in traditional handicrafts [2] and investigating creative teaching in art and design studio classes [3]. Furthermore, studies have evaluated learning methodologies to develop "4C skills" [4]. Norman [5] also proposed modern design education for a more comprehensive, interdisciplinary, and scientifically grounded approach [5]. It has coincided with the need

for those four types of knowledge, as outlined in the 2030 OECD Learning Compass. Meyer & Norman's critique of current design education aligns with the four knowledge types outlined in the OECD 2030 learning framework [6]. The emphasis on disciplinary knowledge, which only includes specific skills and methods, appears narrow-minded and overlooks the importance of interdisciplinary, epistemic, and procedural knowledge. The reform highlights the importance of having a comprehensive, connected, and functional grasp of design, considering real-life challenges from various areas, understanding the essence of design issues, and being skilled in carrying out design procedures, frequently learned through practical experience in the field.

In Taiwan's design education, the traditional approach of teachers developing students' professional competence through the training process of individual courses has also reached a bottleneck and given rise to many practical studies on design education innovation and industry–academia collaboration [7–11]. This situation is mainly due to rapid changes in the design industry over the past two decades [2]. It has resulted in the requirements for designers' professional competency becoming more complex and varied, such as interdisciplinary abilities, social skills [7], team cohesion [11], leadership [12], and entrepreneurship [10]. These competencies are highlighted in the context of sustainable development in traditional handcrafts through a design thinking-based study [2].

Contrastingly, the innovative approach presented in this study seeks to integrate interdisciplinary knowledge with practical design practices. The course is structured into five stages: design proposal, workshop practice, digital design, marketing test, and final marketing. It includes product development, e-commerce marketing, sales, and intellectual property rights protection in collaboration with the 3S4S studio. It is underpinned by a distributed scaffolding framework that supports students' professional growth and interdisciplinary abilities.

Designers with interdisciplinary competence can bridge the gap between different fields of knowledge and integrate diverse perspectives into their design solutions. They can leverage their understanding of the natural sciences, humanities, and design thinking to create sustainable and innovative designs that meet the needs of the present without compromising the future. Social skills are essential for designers to effectively communicate and collaborate with clients, stakeholders, and team members [2,7]. Team cohesion is crucial in design projects that involve collaboration with colleagues from various disciplines. Designers must work cohesively within teams, leveraging their expertise, and effectively co-ordinating efforts to achieve shared goals [11]. Leadership skills enable designers to take charge and guide design projects to success. They can demonstrate initiative, motivate team members, and make critical decisions to ensure the achievement of design objectives [12]. Entrepreneurship plays a vital role in the professional competency of designers, as it fosters an entrepreneurial mindset and the ability to identify opportunities, take calculated risks, and bring innovative design solutions to the market [2,10].

In response to this global challenge, the OECD Learning Compass 2030 report proposes three transformative competencies for students to contribute to our future world: creating new value [13], reconciling tensions and dilemmas, and taking responsibility [14]. The satisfaction of these three transformative competencies will provide a concrete basis for future education towards the vision of knowledge and skills for the 2030 Sustainable Development Goals of the United Nations [1,15,16]. In comparison, the articles mentioned above are closely related to the transformative competencies discussed in the OECD Learning Compass 2030 report [1]. These studies explore various aspects of professional competencies in design, such as innovation competency [7], intercultural teamwork competency [11], leadership competence [12], and entrepreneurship [10], which align with the transformative competencies emphasized in the report. These studies provide empirical evidence and practices for fostering transformative competencies in students, enabling them to address future challenges and contribute to sustainable development [2].

In comparison, the traditional methods appear narrow-minded as they limit the development of interdisciplinary, epistemic, and procedural knowledge. However, the

innovative approach presented in this study offers a comprehensive and connected understanding of design, considering real-life challenges from various areas, and emphasizes skills frequently learned through practical experience, thus addressing the current industry requirements, and fostering competencies that are more aligned with today's dynamic professional landscape.

### 1.2. Research Objectives

This study aims to develop an optimized learning pathway in an interdisciplinary product and media design education curriculum. Through an elaborated distributed scaffolding theory, learning resources in course practice can be fully configured to effectively enhance and fast-track university students' competency and interdisciplinary knowledge in product design and media design. This study's research objectives, questions, and methodology are briefly described below.

1. This study develops an optimized interdisciplinary learning pathway based on distributed scaffolding theory to foster knowledge innovation and competency enhancement for product and media design students. The research question explores how to optimize such pathway design leveraging scaffolding theory. The method implements distributed scaffolding by integrating faculty support systems, enabling effective knowledge flow and student capability cultivation, as evidenced by outstanding project outcomes. This optimized learning pathway will also enable the selection of outstanding student prototypes for mass production [17–20].
2. This study implements a collaborative triple-helix model engaging a ceramics studio to integrate product design, media design, and e-marketing education. The research investigates the potential of this collaborative approach in enhancing interdisciplinary knowledge and honing practical skills for students. The method establishes synergetic university–industry–government interactions, allowing students to achieve innovative design and marketing accomplishments through active participation [21–23].
3. This study examines the professional competencies and interdisciplinary knowledge required to equip product- and media design students for future complexities. The research focuses on developing a curriculum prototype tailored to cultivate transformative competencies using immersive teaching and learning techniques. The method applies distributed scaffolding strategies to provide customized support for capability building, engaging students in projects to expand critical faculties crucial for navigating uncertainties ahead, as outlined in the OECD Learning Compass 2030 report [1,15,24].

## 2. Literature Review

### 2.1. Knowledge Competency

The OECD Learning Compass 2030 report emphasizes the importance of knowledge [1]. This knowledge encompasses facts, concepts, ideas, and theories contributing to our world understanding. It encompasses both theoretical concepts and practical understandings derived from hands-on experience. As education systems evolve, there is a shift towards viewing disciplines as interconnected systems, highlighting the need for interdisciplinary knowledge [15,20]. The report identifies four key types of knowledge: disciplinary, interdisciplinary, epistemic, and procedural. They are briefly described in the following.

1. Disciplinary knowledge encompasses concepts and details within a specific subject, providing a foundation for intrinsic structure and underlying principles. It serves as a basis for further learning, facilitating the exploration of interdisciplinary connections and the practical application of knowledge in different contexts [7,25,26].
2. Interdisciplinary knowledge focuses on understanding and solving complex problems by integrating concepts and content from multiple disciplines. It nurtures valuable competencies, including transferring key concepts, recognizing interconnectedness, thematic learning, creating new topics, and project-based learning [1,27,28].

3. Epistemic knowledge involves understanding subject specialists' and practitioners' thinking and working processes. It encourages students to connect knowledge with real-world problems and fosters the development of "learning to learn" skills [29].
4. Procedural knowledge entails understanding how to perform specific actions or steps to achieve a goal. This transferable knowledge is applicable in various contexts and is valuable in solving complex real-world problems. Design thinking approaches are often employed to apply procedural knowledge effectively [30–34].

### 2.2. Triple-Helix Model

The triple-helix model, which encompasses collaboration between universities, industry, and government, fosters interdisciplinary knowledge innovation [22]. In the era of globalization, this model highlights the increasing intensity and complexity of synergies among these three entities in the knowledge development process. Industries now recognize the value of solid knowledge provided by universities and government R&D departments in a highly competitive global environment. At the same time, universities are expanding their participation in economic development through industry collaborations. This empirical and evidence-based model offers a systematic perspective on interdisciplinary knowledge innovation, facilitating innovative knowledge transfer and incubating new creative industries [21,23]. Moreover, this model illustrates the transformational journey of universities as they contribute to the invention and innovation of industries' economic power [35].

### 2.3. Distributed Scaffolding Theory

Scaffolding theory, inspired by the scaffold metaphor in construction, was developed by Wood [36] to describe how children receive support from mentors to solve problems. Vygotsky [37] introduced the concept of the zone of proximal development (ZPD), which refers to tasks that learners can accomplish with appropriate assistance beyond their current abilities. The combination of the scaffold metaphor and the ZPD concept forms the foundation of scaffolding theory and guides the design and practice of this study [36,38,39].

As learning environments become more complex, distributed scaffolding offers practical guidelines for teachers to select appropriate scaffolding strategies [17–20]. Tabak [17] proposes three scaffolding models within the framework of distributed scaffolding: differentiated scaffolding, redundant scaffolding, and synergistic scaffolding. These scaffolds work together to complete learning tasks.

Differentiated scaffolding supports different learning goals with different characteristics. For example, software scaffolds are suitable to support highly repetitive training activities, and teacher scaffolds play a pivotal role in evaluating the learning process. Redundant scaffolding supports the same learning goals with different types of scaffolds for different learners, and well-prepared redundant scaffolding can effectively reach a more comprehensive range of learners with different ZPDs. Synergistic scaffolding consists of multiple co-occurring and interacting scaffolds supporting the same need. In a collaborative industry–academia project, the supporting scaffolds of disciplinary knowledge provided by school education and the supporting scaffolds of practical expertise from industry are linked through a synergistic strategy to achieve a common goal [17,20].

## 3. Course Design Methodology

### 3.1. Course Description

The curriculum design in this study aimed to integrate disciplinary knowledge in product and media design with interdisciplinary design practices. The course included product development, e-commerce marketing, sales, and intellectual property rights protection in collaboration with Sanshing Four Seasons Celadon Studio (3S4S). In a voluntary partnership with the government, it has successfully nurtured local talent and promoted ceramic art and culture brands from 2017 to 2019 in Yilan County, Taiwan. With the support of these successful experiences, 3S4S has demonstrated great enthusiasm and support for

constantly developing and supporting local talents in creativity and entrepreneurship. This is the main reason for encouraging the research team to seek co-operation to develop talents.

The one-year curriculum ran from August 1, 2019, to July 31, 2020, encompassing two academic semesters. Each semester lasted for 18 weeks, totaling 36 weeks, with students engaging in coursework for three hours weekly. In addition, there was an opportunity for students to partake in a six-day workshop during the winter break between the two semesters, which was dedicated to learning the techniques of clay shaping and kneading. The course involved three teachers specializing in product design, media design, and ceramics, supported by a course assistant.

At the beginning of the course, we will explain to the students that the implementation of this course is due to subsidies planned by the Ministry of Education in Taiwan. Through this, students can gain opportunities and resources for industry and school to cultivate talents jointly. All students in the course expressed their willingness to participate in this project.

The course was open to second-year students, with 17 students enrolling in the first semester and 15 in the second. Participation was organized into five groups, with varying levels of engagement as follows:

1.  Group A, consisting of four members, was fully engaged during the first semester. However, due to the perceived complexity of their project, the group members decided not to continue into the second semester.
2.  Group B, comprising four members, participated completely in the first semester. Yet, due to the considerable distance from the school to the 3S4S workshop, three members chose not to enroll for the second semester.
3.  Groups C and D, with five and four members, respectively, completed both semesters of the course without interruption.
4.  Group E, with five members, did not participate in the first semester. Motivated by the influence of their peers and personal interests, they proactively developed and created their project ideas during the winter break. Their initiative received recognition and encouragement from the instructor, leading them to complete the winter workshop and the second semester actively.

In summary, there were 17 students in semester one and 15 in semester two. The group work fostered peer learning and creative development through a flexible structure accommodating individual needs.

The course was structured into five stages: (1) design proposal, (2) workshop practice, (3) digital design, (4) marketing test, and (5) marketing. The first two stages were conducted in the first semester, the next two in the second semester, and the final stage as a continuation of the course.

*3.2. Course Planning*

The course consisted of heterogeneous groups with 4–5 students per group, combining product design and media design students. The process of the ceramic creative product course practiced by the five groups is illustrated in Figure 1. The green circles represent the successful completion of the learning tasks, while the orange circles represent groups experiencing difficulties but open to teacher suggestions. Red circles represent groups that face challenges and are unwilling to modify their tasks. The course was divided into five stages, summarized below.

1.  **Design proposal**. Each group presented four proposals related to the Mother's Day theme, including story proposals, sketches, design plans, and mold prototypes. The proposals underwent multiple revisions based on teacher evaluations until approval was granted.
2.  **Workshop practice**. The approved proposals were brought to life in the ceramic studio, where the students participated in hands-on activities such as kneading, shaping, modifying forms, repairing, biscuit firing, glazing, painting, and final firing. Microproduction techniques were also explored.

3.   **Digital design**. Students used digital tools to complete tasks such as product photography, brand logo design, image animation, poster creation, and cool card design.
4.   **Marketing test**. The products were showcased to the public through a Christmas Creative Fair, accompanied by a questionnaire survey. The 3S4S manager evaluated and approved selected products for presentation on the Pinkoi Online Design Gallery website.
5.   **Marketing**. Limited product production and patents were obtained to protect intellectual property. The patent aimed to license the aesthetic value of the "Funny Monkey" artwork, offering an experiential understanding of its economic value.

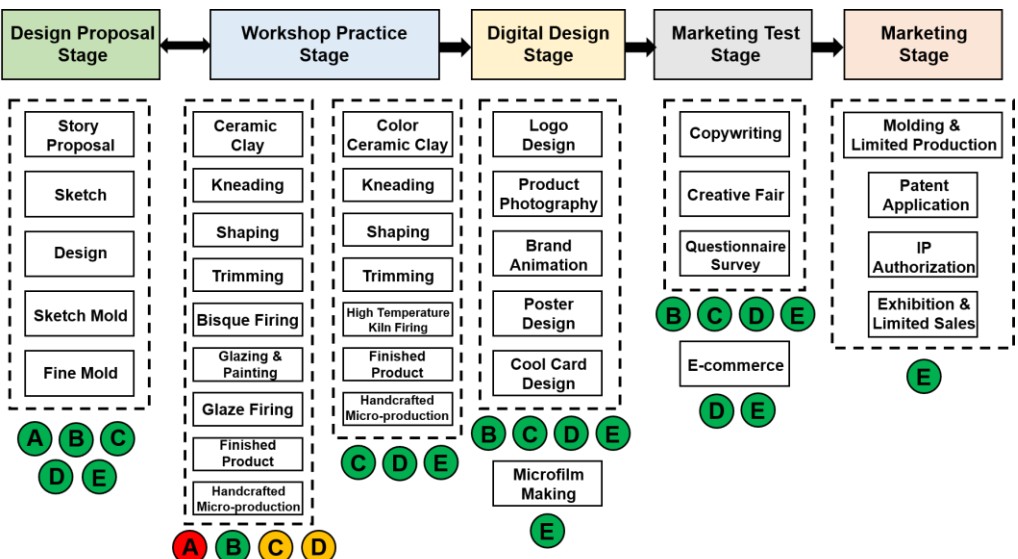

**Figure 1.** A roadmap of course practice and group implementation. The green circles represent the successful completion of the learning tasks, while the orange circles represent groups experiencing difficulties but open to teacher suggestions. Red circles represent groups that face challenges and are unwilling to modify their tasks. The letters A–E depicts the name of groups respectively.

*3.3. Course Practice*

3.3.1. Design Proposal Stage

In the initial stage (Figure 1), students embarked on an interdisciplinary design proposal focused on developing filial piety and cultural and creative products for Mother's Day. This stage aligned with the principles of knowledge for 2030 [1,15], allowing students to transfer key concepts [27] and explore the interconnectedness of different disciplines through thematic learning [40] and project-based approaches [41]. The teacher provided comprehensive guidance and support during this stage using redundant scaffolding techniques [17,20]. Five design proposals were approved, including the Health Tableware Group (Group A), Flower Tea Plate Group (Group B), Little Elephant Chopstick Holder Group (Group C), Spinning Tableware Holder Group (Group D), and Funny Monkey Group (Group E). Each group's design concept or brand story is briefly described below.

1.   Group A designed the Health Tableware, incorporating various sports and wellness designs to promote the idea that expectant mothers want healthy bodies. This included condiment jars and tableware inspired by yoga and hanging kanas.
2.   Group B created the Flower Tea Plate, a celadon tableware with a simple hand-painted pattern of flower petals, symbolizing Mother's Day. The finished product could be used to serve cakes and desserts or as a display of small accessories such as earrings and rings.

3.  Group C designed the Little Elephant Chopstick Holder, representing the special bond between mother and child. The design reflects the close relationship between parents and children, with the elephant symbolizing this extraordinary connection.
4.  Group D developed the Spinning Tableware Holder, a delicate spiral cutlery holder that securely holds cutlery, representing the mother's love for her child.
5.  Group E designed the Funny Monkey, a figure with big ears that symbolizes listening to everything in the world. The different gestures of the figure, including folded hands, kneeling down, and both hands in prayer, represent blessings, giving, care, and love, similar to a mother's unconditional love for her child.

### 3.3.2. Workshop Practice Stage

Refer to Figure 1. The second stage of the handicraft practice results of the workshop is briefly described below.

1.  Group A completed the firing process for their first batch of work. However, a disagreement during the sketch proposal stage resulted in a loss of interest among group members, leading them to not continue with the course in the following term.
2.  Groups B and E followed the curriculum, gradually advancing from clay kneading to kiln production.
3.  Groups C and D encountered two challenges in their initial work. First, they faced difficulties in mastering the shape kneading. Second, they needed help with the quality of the glaze lines. Despite repeated practice, the workshop teachers did not approve of their work. As a result, they went back to the first stage and created new proposals. Group C, in particular, shifted its focus from designing an ornament storage box to a chopstick holder group. Likewise, Group D transitioned from designing a complete tableware set for afternoon tea to a tableware stand. These refinements allowed both groups to develop transformative competencies, such as reconciling tensions and dilemmas [7,15] and taking responsibility [14] for their work.

### 3.3.3. Digital Design Stage

In the third stage (Figure 1), the students engaged in digital design, using various digital tools as differentiated scaffolding [17]. This included using digital photography, graphic and animation software, and digital editing software to design brands. Synergistic scaffolding was also employed, incorporating the development of marketing strategies, logo design to convey brand concepts, animated content to enhance brand image, and the creation of marketing posters and promotional materials. Figure 2 presents the results of the digital design phase for each group, featuring logo designs, product photography, and marketing posters.

### 3.3.4. Marketing Test Stage

In the fourth stage, the marketing test, students engaged in activities such as photography and copywriting to compete in selling their products on the e-commerce platform "Pinkoi Design Gallery". This stage allowed students to enhance their graphic integration skills. Group E also documented their learning process using microfilm to effectively convey the complete design concept of cultural and creative product development. Following a successful Creative Fair (Figure 3a,b), the industry-academic partner, 3S4S, recognized the Spinning Tableware Holder (Group D) and the Funny Monkey (Group E) as the best products. These products were made available for sale at the Pinkoi Design Gallery (Figure 3c,d). The limited production of the Funny Monkey (Group E) quickly gained popularity among regular 3S4S customers, resulting in a sold-out status shortly after its launch. In particular, the participation of the 3S4S manager served as a form of redundant scaffolding [17,20] in supporting and guiding students during the marketing stage. Significantly, participation effectively strengthened and improved the market value of the product. Furthermore, the 3S4S manager played a crucial role in transferring innovative knowledge and fostering the growth of new creative industries within the triple-helix model [23].

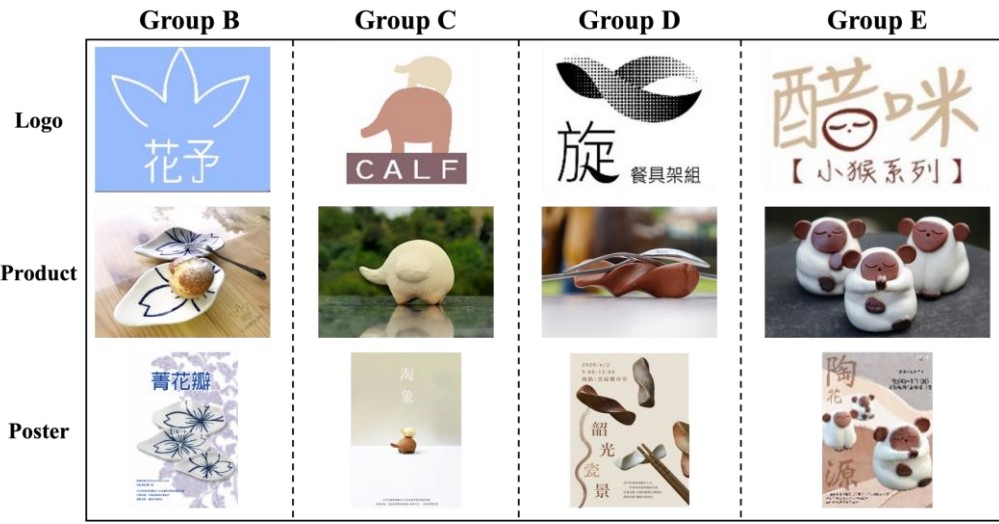

**Figure 2.** Results of the digital design of each group.

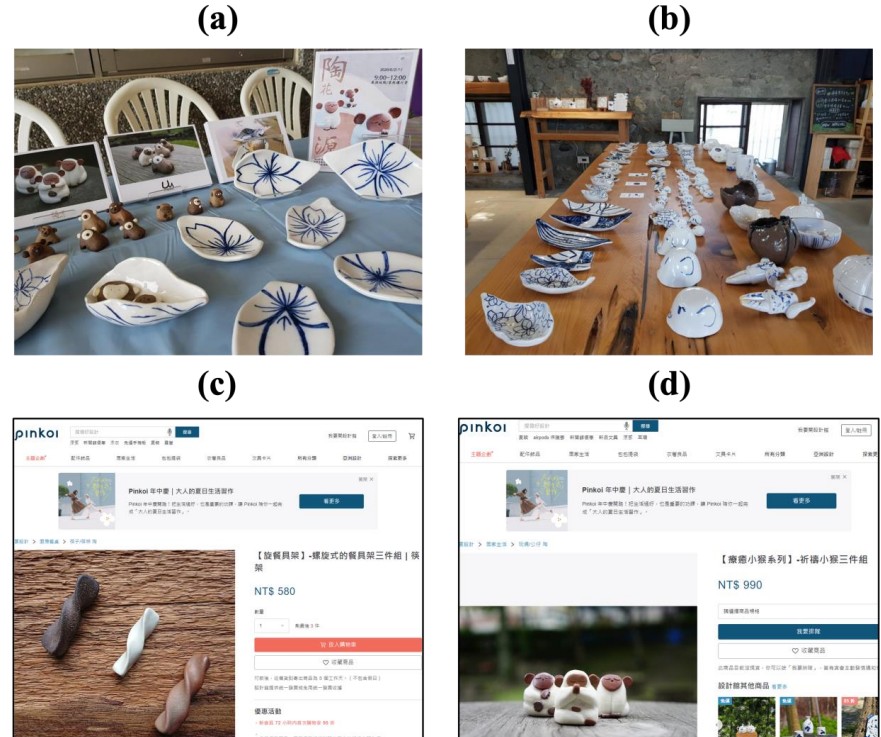

**Figure 3.** Creative Fair and Online Marketplace for Student Products. (**a**) Photograph of products and digital design of groups B and E, (**b**) photograph of exhibition of products of all groups, (**c**) Pinkoi Design Gallery pages of group D and (**d**) group E respectively.

### 3.3.5. Marketing Stage

In the study's later stage, feedback on the students' work was collected through a creative market questionnaire (Figure 1). Respondents expressed optimism about the cultural and creative products, describing them as cute, sophisticated, healing, and unique. They also appreciated the decorative value and practicality in daily life. This feedback provided a clear direction for developing the "Funny Monkey" mold. Group E incorporated the suggestion of adding a money box function to their design (Figure 4). With the guidance of market feedback, the team's improved cultural and creative product received a design patent in Taiwan in May 2021. This notable achievement underscores the power of interdisciplinary learning in this study, particularly in fostering professional and transformative

competencies among students. These competencies include creating new value, fostering team cohesion, promoting entrepreneurship, and cultivating a sense of responsibility and objectives that align with the course design goals [7,8,10–15].

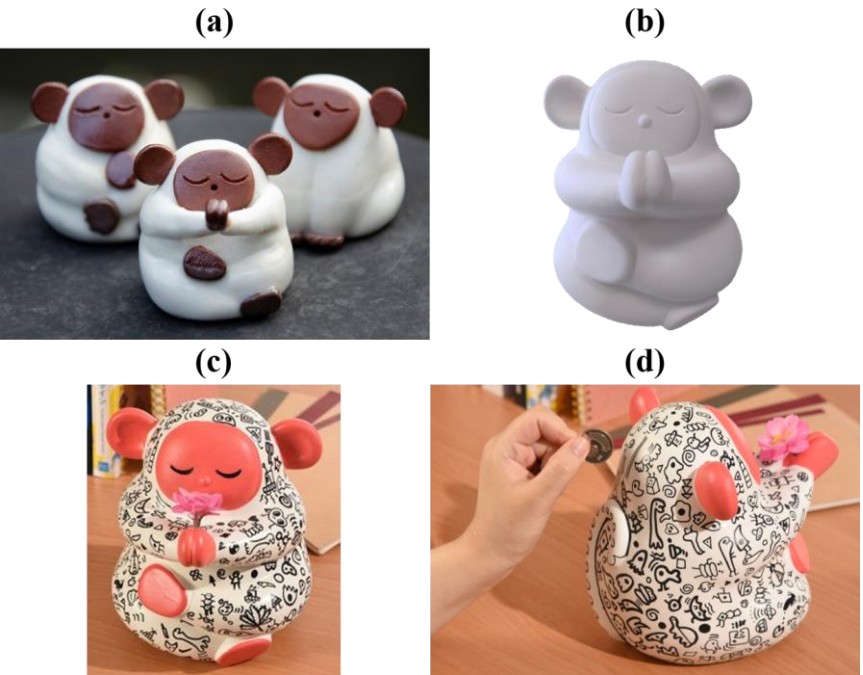

**Figure 4.** Funny monkey ceramic creative products. (**a**) prototype, (**b**) figure mold, (**c**) painted figure, (**d**) figure money box.

### 3.4. Course Scaffolding Framework

This study unleashes students' design potential and fosters interdisciplinary knowledge through practical design practice, heterogeneous grouping, and industry–academic collaboration. The study improved professional competencies by constructing a distributed scaffolding framework [20] and expanded the scope of interdisciplinary knowledge innovation [7].

Scaffolding, which includes the support of humans and artefacts, plays a crucial role in this study [17,18,42]. Human scaffolding includes group peers (PS1, PS2, PS3, PS4, PS5), media design teachers (T1), product design teachers (T2), ceramic crafts teachers (T3), patent offices (I1), cultural and creative product distributors (I2), and mold developers (I3). Artefact scaffolding involves computer graphics, audiovisual software (A1), ceramic workshop software, and hardware (A2). The course encompasses five learning objectives: completing a design proposal (O1), workshop activities (O2), digital design tasks (O3), marketing tests (O4), and planning for continuous cultural and creative marketing (O5).

Figure 5 provides a schematic representation of the interdisciplinary curriculum based on the distributed scaffolding perspective derived from the group implementation results discussed in Section 3.2. It is important to note that the blue arrows represent differentiated scaffolding, the green arrow represents redundant scaffolding, and the orange arrow represents synergistic scaffolding.

In Figure 5, the progression of each group's learning state toward different objectives (O1 to O5) is driven by differentiated scaffolding. This is facilitated through a variety of scaffolding resources, including peer scaffolds (P1 to P5), academic scaffolds (T1, T2), industry scaffolds (T3, I1 to I3), and artefact scaffolds (A1 and A2). It is important to note that the interdisciplinary teachers (T1, T2, and T3) collaborated to establish clear objectives for each stage of the course. Additionally, the redundant scaffolding provided by academic scaffolds (T1 and T2) was crucial in supporting students, resolving problems, and ensuring curriculum progress. Furthermore, synergistic scaffolding, co-constructed by

the academic and industry scaffolds, facilitated the transition from in-house prototypes to real-world products.

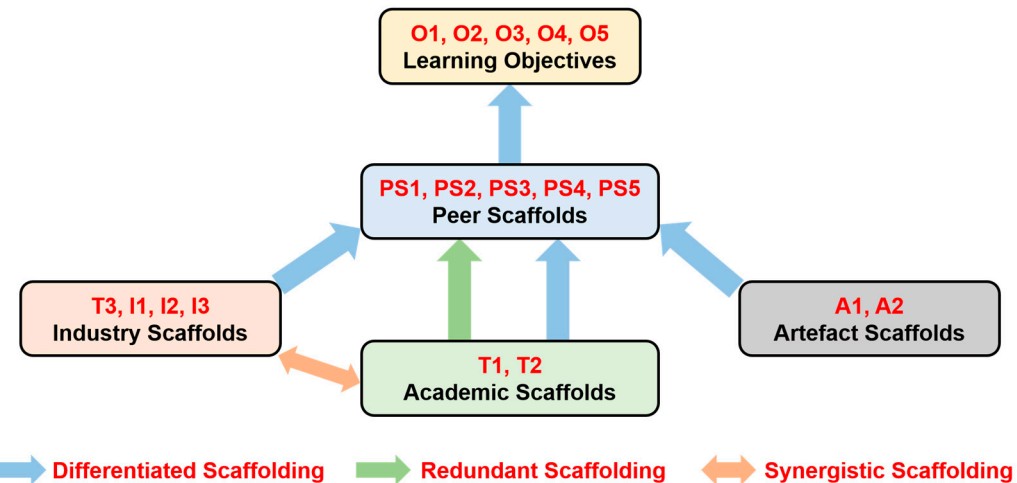

**Figure 5.** Schematic of distributed scaffolding for practice in the interdisciplinary curriculum.

## 4. Results and Discussion

### 4.1. Questionnaire Analysis

The present study conducted anonymous questionnaire surveys among enrolled students at the end of each semester. The questionnaire, which consists of 17 questions, was provided by the Digital Humanities Project Office of the Ministry of Education of Taiwan. Using a Likert five-point scale, the survey investigated three aspects of the students' course learning: (1) actual feelings about the course, (2) improvement in abilities and knowledge resulting from the course, and (3) satisfaction with course expectations. In the first semester (Semester 1), 14 respondents completed the questionnaire (response rate: 82.35%), while in the second semester (Semester 2), 15 respondents completed the questionnaire (response rate: 100%). Among them, nine students took courses in the first and second semesters.

Table 1 shows the analysis results of the course questionnaire. The mean values indicate the average ratings given by the students, while the standard deviations represent the variability in their responses. In Aspect 1, which focuses on students' feelings about the course, all questions received scores higher than the average value on the five-point scale, indicating that the students generally agreed with the descriptions. The mean values in Semester 2 were consistently higher than in Semester 1, suggesting that the overall learning outcomes were better in the second semester.

For Aspect 2, which refers to improving abilities and knowledge, the mean values for all questions were above average, indicating positive outcomes. Once again, Semester 2 exhibited higher mean values than Semester 1, indicating a more favorable improvement in the second semester. Regarding Aspect 3, which focuses on course expectations, Semester 2 reported a much higher mean value than Semester 1, indicating a high satisfaction with the course expectations in the second semester.

In general, analysis of the responses to the questionnaire reveals that the students enrolled had positive perceptions of the course and experienced improvements in their abilities and knowledge. Additionally, the second semester demonstrated higher mean values, indicating better overall course learning outcomes than the first.

It is worth noting that the standard deviations in Semester 2 were consistently more minor than those in Semester 1. This suggests that the enrolled students in the second semester had more consistent opinions regarding the effectiveness of the course in improving their abilities and knowledge.

The questionnaire analysis shows that the learners clearly understood the course, its objectives, and its content (Aspect 1). Additionally, the enrolled students developed

interdisciplinary abilities (Questions 1-4, 2-3), social skills (Questions 1-5, 2-2, 2-6), team cohesion (Questions 2-2, 2-3), leadership (Questions 1-6, 2-2, 2-3), and entrepreneurship (Questions 1-6, 2-5, 2-6) through the course's guided distributed scaffolding. These findings align with previous studies [7,10–12], highlighting the importance of practical experiences, industry collaboration, and the application of digital humanities in enhancing learners' professional and interdisciplinary competencies.

**Table 1.** Analysis of Course Questionnaire.

| Aspect 1: Actual Feelings about the Course | | | |
|---|---|---|---|
| **Question** | **Semester** | **Mean** | **Standard Deviation** |
| 1-1 Clear emphasis on course description | 1 | 3.786 | 0.802 |
| | 2 | 4.625 | 0.500 |
| 1-2 Sequential and progressive course planning | 1 | 3.929 | 0.730 |
| | 2 | 4.688 | 0.479 |
| 1-3 Course content aligned with objectives | 1 | 4.000 | 0.784 |
| | 2 | 4.625 | 0.500 |
| 1-4 Interdisciplinary knowledge transmission | 1 | 3.929 | 0.829 |
| | 2 | 4.688 | 0.479 |
| 1-5 Appropriate use of on-line platforms for enhanced teacher-student interaction | 1 | 3.929 | 0.917 |
| | 2 | 4.500 | 0.816 |
| 1-6 Industry-academia collaboration | 1 | 4.286 | 0.726 |
| | 2 | 4.938 | 0.250 |
| **Aspect 2: Improvement in Abilities and Knowledge** | | | |
| 2-1 Application of digital tools for data processing and analysis | 1 | 3.929 | 0.997 |
| | 2 | 4.563 | 0.727 |
| 2-2 Enhancement of collaboration with individuals of different professional backgrounds | 1 | 4.286 | 0.726 |
| | 2 | 4.750 | 0.577 |
| 2-3 Improvement in interdisciplinary learning and application | 1 | 4.286 | 0.726 |
| | 2 | 4.688 | 0.602 |
| 2-4 Increase in interest in proactive exploration in the field of digital humanities | 1 | 4.000 | 0.877 |
| | 2 | 4.625 | 0.619 |
| 2-5 Enhancement of abilities in applying digital humanities to practical contexts | 1 | 3.929 | 0.829 |
| | 2 | 4.813 | 0.403 |
| 2-6 Increase in confidence for future employment | 1 | 4.143 | 0.949 |
| | 2 | 4.375 | 0.885 |
| 2-7 Willingness to recommend the course to other students | 1 | 4.071 | 0.730 |
| | 2 | 4.500 | 0.816 |
| 2-8 Will to take advanced courses | 1 | 4.071 | 0.730 |
| | 2 | 4.500 | 0.816 |
| **Aspect 3: Satisfaction with Course Expectations** | | | |
| 3-1 The course has met the course expectations compared to before enrollment | 1 | 3.929 | 0.730 |
| | 2 | 4.625 | 0.500 |

Furthermore, the expected results of the course, as documented through student interviews, included three specific accomplishments. First, the students gained practical knowledge of the ceramic industry by creating original works and transitioning to microscale production. This indicates that the course provided them with hands-on experience and insights into the practical aspects of the ceramic industry.

Second, they familiarized themselves with the operational models of interdisciplinary industries through a series of media design integration courses related to microscale production

products. The course provided opportunities for students to develop a holistic understanding of the industry's functioning, incorporating various disciplines and approaches.

Lastly, the students developed a transferable model of interdisciplinary knowledge through the practical implementation of the course. This suggests that the course facilitated the development of skills and competencies that can be applied in different domains and disciplines. By participating in collaborative projects and problem-solving activities, students gained valuable experience in teamwork, critical thinking, and effective communication.

In summary, the questionnaire analysis and student feedback results indicate that the course achieved its expected results. It provided students with practical knowledge, interdisciplinary skills, and the ability to transfer their learning to different contexts. These results demonstrate the effectiveness of the course in fostering professional growth, interdisciplinary abilities, and the application of digital humanities knowledge.

### 4.2. Transformative Competencies Development by Distributed Scaffolding

This section examines the application of distributed scaffolding theory in the course, focusing on the design of cultural and creative items related to Mother's Day. The key findings related to transformative competencies and the impact of scaffolding resources on student learning are summarized as follows:

1. Teacher scaffolding (T1, T2, and T3) was crucial in supporting students and ensuring progress [8]. Successful outcomes were facilitated through collaboration between teachers and students, while conflicts within groups and a reluctance to accept teacher guidance hindered progress. As shown in Figure 1, Group B showed a positive impact of teacher support. The only student in the group who continued participating in the second semester of the course successfully achieved the goal with the teacher's help. In contrast, Group A faced difficulties due to conflicting ideas and a reluctance to accept teacher co-ordination. Team cohesion and social skills were essential in design projects [7,11].
2. Students rely heavily on artefact scaffolding tools (A1 and A2) to complete tasks and achieve learning objectives (O1 to O5).
3. The flexible support mechanism of distributed scaffolding significantly contributed to the creative process and the development of transformative competencies such as reconciling tensions and taking responsibility [14,15]. Peer suggestions, teacher responses, and collaboration were integral to the success of distributed scaffolding.
4. With prior knowledge and experience in ceramic workshops, Group E exhibited a higher level of detail and integrity in their work to successfully drive them from the marketing test stage to the marketing stage (as shown in Figure 1). They served as a supportive and inspiring role model for other groups.
5. During the marketing stage, the teacher scaffolding expanded to include patent offices (I1), distributors (I2), and mold developers (I3), providing invaluable support for the successful sales of Group E's work on the Pinkoi e-commerce platform.

These findings align with the concept of transformative competence, emphasizing the role of teacher scaffolding in addressing tensions and dilemmas faced by students [8]. It provides evidence supporting the emphasis on transformative competencies for future talent in the 2030 OECD Learning Compass. Peer scaffolding within the curriculum framework stimulates students' creativity, enabling them to overcome challenges, create new values, and develop a sense of responsibility [12–14].

## 5. Conclusions

### 5.1. Optimised Learning Pathway for Interdisciplinary Curriculum

One of the significant contributions of this research is the development of an optimized learning pathway in interdisciplinary design education. The curriculum design framework successfully establishes a pathway for knowledge innovation and competency development by implementing distributed scaffolding theory and integrating interdisciplinary faculty

support systems. The following points highlight the essential findings and contributions of this study.

1.  This study successfully explores an optimized learning pathway for interdisciplinary knowledge innovation in designing professional practice courses (OECD, 2019). It implements distributed scaffolding theory [17,18,20,42] to enrich the pedagogy of product and media design courses. The curriculum design framework integrates interdisciplinary faculty support systems for product design, media design, and e-commerce marketing in collaboration with a ceramic workshop involved in regional development.
2.  The curriculum design framework effectively establishes knowledge innovation derived from the triple helix structure of university–industry–government [23,35]. The students in the course achieve significant results, including innovative ceramic designs, digital media marketing, branding, and online store operations, indicating their acquisition of interdisciplinary knowledge and transformative competencies.
3.  Scaffolding theory provides a robust support structure for students, as illustrated in Figure 1. Teacher support, peer collaboration, and the synergy between hardware and software are essential. Teacher reflections and strategies for addressing student challenges contribute to improving students' learning paths, as shown in Groups C and D in Figure 1.
4.  Effective peer scaffolding drives motivation and achievement in interdisciplinary learning. Conflicting ideas from diverse backgrounds can pose challenges but can be overcome through guidance and mutual understanding among peers [7,8,11].

Overall, this study demonstrates the effectiveness of the optimized learning pathway and the role of scaffolding theory in promoting knowledge innovation and interdisciplinary competence development in design education.

### 5.2. Transformative Competencies Development

One of the main contributions of this research is the promotion of transformative competencies through the development of an interdisciplinary curriculum. The curriculum design successfully establishes a roadmap for fostering knowledge innovation and competence among students by implementing distributed scaffolding theory and integrating interdisciplinary faculty support systems. The following points highlight the essential findings and contributions of this study.

1.  The curriculum focuses on building foundational disciplinary knowledge in product and media design. These initial stages give students the technical details and basic knowledge of their respective fields. They serve as a solid foundation for further exploration of interdisciplinary knowledge [1].
2.  The subsequent stages of the curriculum, the marketing test, and the marketing stages encompass epistemic, procedural, and interdisciplinary knowledge. These stages aim to develop students' critical thinking, strategic action, and problem-solving abilities [12]. Through engagement in the entrepreneurial process, some participating students completed real-world projects, obtained patents, and received recognition in creative competitions, showcasing the effectiveness of interdisciplinary collaboration [10,23,35].
3.  Implementing distributed scaffolding theory provides a practical roadmap for designing interdisciplinary curricula. Peer scaffolds, teacher scaffolds, and artefact scaffolds form a cohesive support system that guides students' learning pathways. Teacher companionship is identified as a crucial element in successful student scaffolding. The results highlight students' enhanced learning abilities, creativity, interpretation, and professional competencies through participation in design projects [7,9,11,12].
4.  This research emphasizes the importance of transformative competencies in preparing students for an ever-changing world. By providing tangible evidence and a transferable pedagogical prototype, this study contributes to interdisciplinary education. It

supports the development of individuals equipped with the skills needed to thrive in the future [14,15,33].

Notably, to provide a convincing conclusion based on the questionnaire analysis with the small sample size, this study also focuses on qualitative insights, detailed case studies, and individual testimonies, as presented in Sections 3.3 and 4.2. Those results demonstrate the course's impact. The consistency of improvements across the small sample could be highlighted, and if the results are particularly strong, even a small sample can suggest a positive trend. Triangulating these results with other evaluation forms, such as peer assessments or interviews, teacher observations and evaluations, and project outcomes, such as the Creative Fair showcases, can help reinforce the questionnaires' findings.

In conclusion, this research has successfully developed an interdisciplinary curriculum that promotes knowledge innovation and fosters transformative competencies among students. By integrating distributed scaffolding theory and engaging students in collaborative design projects, the curriculum effectively builds foundational disciplinary knowledge, develops critical thinking and problem-solving skills, and empowers students to navigate real-world challenges. This research provides a practical roadmap for designing interdisciplinary curricula. It emphasizes the importance of transformative competencies in preparing students for the future (as shown in Figure 1). The findings contribute to the field of interdisciplinary education and highlight the importance of teacher scaffolding, peer collaboration, and the integration of diverse scaffolding resources in enhancing student learning outcomes and professional competencies (as shown in Figure 5).

**Author Contributions:** Conceptualization, Y.-F.K.; methodology, Y.-F.K. and H.-C.C.; course design and teaching practice, Y.-F.K.; formal analysis, Y.-F.K., H.-C.C. and J.-H.L.; investigation, Y.-F.K.; writing—original draft preparation, Y.-F.K. and H.-C.C.; writing—review and editing, Y.-F.K., H.-C.C. and J.-H.L.; visualization, Y.-F.K. and H.-C.C.; funding acquisition, Y.-F.K.; project administration, Y.-F.K. All authors have read and agreed to the published version of the manuscript.

**Funding:** This research was funded by the National Science and Technology Council, ROC, under contract numbers. MOST-108-2410-H-431-007 and MOST 109-2420-H-431-001-MY3. It was also supported by the Ministry of Education of Taiwan under contract number MOE-108-1-A28 and Huanggang Normal University, China under contract number 2042022027.

**Institutional Review Board Statement:** Not applicable.

**Informed Consent Statement:** Not applicable.

**Data Availability Statement:** Data are unavailable due to privacy or ethical restrictions.

**Acknowledgments:** Y.F.K. would like to thank Da-Chung Wu for his help in teaching the product design course and 3S4S for their full support in teaching ceramics and e-commerce practices. We would also like to thank the Ministry of Culture of Taiwan's city brand, "JINHO", in Yilan County, for supporting and assisting with this study.

**Conflicts of Interest:** The authors declare no conflict of interest.

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
