# Peer review of "Exploring an Interdisciplinary Curriculum in Product and Media Design Education: Knowledge Innovation and Competency Development"

_sustainability, doi:10.3390/su152316369_

Round 1

Reviewer 1 Report

Comments and Suggestions for Authors

This study examines the impact of an interdisciplinary curriculum in product and media design education, which incorporates workshops, digital design, and marketing activities guided by teachers and mentors. The study attempts to show that this curriculum promotes knowledge innovation and enhances students' professional skills, creativity, and problem-solving abilities. The study emphasises the importance of scaffolded support systems that contribute to transformational competencies. The study offers practical guidance for teaching strategies and educational innovation.

This paper is a very interesting read. The highlight is the discussion of the student projects and the diagrams that clearly illustrate the concepts created by the students, and provide concrete examples of the course design in action and some proof that it worked, in the first iteration.

The paper has a great introduction outlining the background for the work, and it also has a good literature review section that fully explains the theory behind the approach. What is possibly missing from the introduction/literature review is a discussion on how this type of course has been taught before, so that a comparison can be made between a traditional approach and the innovativeness of this approach.

While the introduction outlines a three point research purpose for the paper, it does not have research questions or outline a methodology to address three points in the research purpose. The paper also doesn't return to clearly address the three points in the research purpose in the discussion or conclusion.

The design of the course is nicely explained, and it is well related to the case study from 2019/2020. But it is not clear what are the essential features of the design and how they are related to literature that would enable the reader to reproduce the experience in their own context. The explanation is too closely related to the case study.

In the course design section, the description on lines 180-187 is a bit confusing. The years for 3S4S do not match the years of the course, so the relationship is not clear. The 36 weeks and then later discusses 2 semesters and winter holidays. Can this section be made clearer. Is it a continuous set of courses, meant to be the same students doing a series of courses?

The case study spoke about and the number of students and the number of students in the groups seems a bit inconsistent. Line 193 says 4-5 students per group, there were 5 groups, but only 17 students. The maths isn’t working. It then says 11 regular participants does that meant 11 students were the same between semester 1 and semester 2, or that each semester only 11 students participated. Did the groups change from Sem1 to sem2, were different students working on the same products from semester to semester?

The analysis on the success or otherwise consisted of a student survey, conducted across the two semesters. There is no comparison to previous approaches, or outcomes for other approaches. Only internal analysis comparing one semester to another, that concludes the learning is deeper in the second semester, which I believe should be the expected result.

Section 4.2 seems to be out of place. I felt it was discussing the course design and possibly belonged earlier in the paper.

As discussed above the discussion and conclusion do not revisit the purpose of the paper, so the reader is left asking what are the key takeaways from this experience and how has the paper reached the conclusions it has made.

A few grammatical errors:

Line 140 fosters fostering

Line 260 should that be group C not D

Line 274 in fig 2, should the columns be labelled b-e to match the groups described in previous section

Line 426 prior knowledge indicates groups weren’t equal? This indicates that the success of group E might not relate just to the experience in this course.

Line 416 “a student” or the entire group - goes back to the query about group size and number of students.

Reviewer 2 Report

Comments and Suggestions for Authors

This study aims to develop an optimized learning pathway in an interdisciplinary curriculum for product and media design education. The research uses the lenses of the scaffolding theory to: develop an optimized learning roadmap for enabling interdisciplinary knowledge; implement a triple-helix model to integrate product design, media design, and e-marketing learning; examine the professional competencies and interdisciplinary knowledge required for university students in product and media design. 

The research is well documented with respect to scientific literature however, there are some issues that need to be adressed:

- sample size for the questionnaire research stage is very small (14 students for the first semester, 15 students for the second semester). Although the response rate is 82% and 100% respectively, I believe that the sample size must be increased to draw relevant conclusions. The sample size can be increased as follows: distribute the questionnaire to multiple generations of students or to alumni

- figure 3b should be removed (contains pictures of individuals)

- the authors should explain how the consent of the participants in the study was obtained. 

Round 2

Reviewer 1 Report

Comments and Suggestions for Authors

The authors have done a great job of responding to the feedback, especially in the first half of the paper. The paper still lacks the depth in the conclusions and applicability by others, but it is worthy of publication as readers in this space will get something from the paper.

Reviewer 2 Report

Comments and Suggestions for Authors

The authors addressed all my comments. It is understood now that the sample size can't be increased due to the uniqueness of the program. I believe the paper is suitable for publication.